# Evaluation of Sputtering Processes in Strontium Iridate Thin Films

**DOI:** 10.3390/nano14030242

**Published:** 2024-01-23

**Authors:** Víctor Fuentes, Lluis Balcells, Zorica Konstantinović, Benjamín Martínez, Alberto Pomar

**Affiliations:** 1Instituto de Ciencia de Materiales de Barcelona, ICMAB-CSIC, Campus Universitario UAB, 08193 Bellaterra, Spain; vfuentes@icmab.es (V.F.); balcells@icmab.es (L.B.); benjamin@icmab.es (B.M.); 2Center for Solid State Physics and New Materials, Institute of Physics Belgrade, University of Belgrade, Pregrevica 118, 11080 Belgrade, Serbia

**Keywords:** spin–orbit coupling, epitaxial thin films, complex oxides, growth mechanisms, magnetron sputtering

## Abstract

The growth of epitaxial thin films from the Ruddlesden–Popper series of strontium iridates by magnetron sputtering is analyzed. It was found that, even using a non-stoichiometric target, the films formed under various conditions were consistently of the perovskite-like n = ∞ SrIrO_3_ phase, with no evidence of other RP series phases. A detailed inspection of the temperature–oxygen phase diagram underscored that kinetics mechanisms prevail over thermodynamics considerations. The analysis of the angular distribution of sputtered iridium and strontium species indicated clearly different spatial distribution patterns. Additionally, significant backsputtering was detected at elevated temperatures. Thus, it is assumed that the interplay between these two kinetic phenomena is at the origin of the preferential nucleation of the SrIrO_3_ phase. In addition, strategies for controlling cation stoichiometry off-axis have also been explored. Finally, the long-term stability of the films has been demonstrated.

## 1. Introduction

Oxide thin films, with their unique properties and versatility, stand as catalysts in the pursuit of transformative solutions for electronic and energy applications [1,2]. In addition to the inherent tunability of the physical properties of these oxides in their bulk form, thin film allows the design of emerging functionalities through engineering defect structures, interfaces or even new phase stabilization [3,4]. This last strategy, named epitaxial stabilization, is a versatile tool that has been widely used to enlarge the range and capabilities of functional materials [5]. Taking advantage of the large non-equilibrium conditions used in physical vapor deposition techniques, such as, for example, pulsed laser deposition or magnetron sputtering, oxide thin films of thermodynamically unstable phases have been successfully grown. Furthermore, kinetic effects can be enhanced with the right choice of a substrate that superimposes an additional surface energy to the delicate balance required for grain nucleation [3]. A very-well-studied example in the last years corresponds to the Ruddlesden–Popper (RP) iridates. Each member of the RP family (Sr_n+1_Ir_n_O_3n+1_) is a superlattice of n perovskite layers (SrIrO_3_) separated by rock salt layers (SrO). The presence of heavy 5d iridium atoms gives rise to a strong spin–orbit coupling (~0.5 eV) that is comparable to electronic correlations [6]. As a consequence, a plethora of interesting aspects in physics has been observed, as the electronic properties range from insulating states for the n = 1 (Sr_2_IrO_4_) or n = 2 (Sr_3_Ir_2_O_7_) phases to the semi-metallic behavior of the n = ∞ (SrIrO_3_) phase. Of particular interest is the case of perovskite-like SrIrO_3_ (SIO-113 in what follows), which, being close to a metal-insulator transition, has been suggested as a potential candidate for neuromorphic [7] or catalytic [8] applications or as an active ingredient in resistive-based memories [9,10]. Furthermore, the literature about unexpected SIO-113 electronic properties, including strange metallicity [11,12] or the anomalous or topological Hall effect, is large [13,14,15]. On the other hand, due to its large spin–orbit coupling, it has also been employed in heterostructures for spin torque or spin-to-charge conversion experiments [16,17,18,19,20].

From the point of view of thin-film growth, SIO-113 is a fascinating case in the RP family; while in bulk form, the orthorhombic perovskite phase only stabilizes at high pressures (the stable phase at ambient pressures being hexagonal) [21,22,23], in the thin-film form it has been routinely synthesized on different substrates. This epitaxially stabilized phase is usually treated as a pseudo-cubic structure with a cell parameter of ~3.95 Å. However, it is known that the complex superlattice structure of RP phases (with the alternating perovskite–rock salt stacking) makes the occurrence of secondary phase intergrowths a real challenge for controlling cation stoichiometry and film disorder. Early works have demonstrated that, by using pulsed laser deposition (PLD), thin films of different RP phases (n = 1, n = 2 and n = ∞) can be grown using the very same single target, just by adjusting thermodynamic conditions (oxygen pressure and temperature) during deposition [24,25,26]. It should be noted that, in those works, several stoichiometric target compositions were used (n = 1 and n = ∞), leading to similar results. In contrast, reports on films grown by magnetron sputtering are scarce. Nevertheless, the few existing reports show that the n = ∞ phase is always obtained, even when using off-stoichiometric targets, with no signs of other RP members [9,10,15,27,28]. These results evidence that kinetic effects during growth, even those that are ignored sometimes, should be taken into consideration when analyzing epitaxial growth and stability. Research on this topic is vital because the sputtering process has intrinsic benefits over PLD for a wide area of development and scalability up to industrial capabilities.

In this study, we present a novel analysis of the growth behavior of epitaxial SIO-113 thin films fabricated via magnetron sputtering. Our results uncover a remarkable tendency for the perovskite-like phase to nucleate, contrary to traditional thermodynamic predictions. By conducting thorough microstructural examinations, we make evident for the first time the pivotal role of backsputtering processes and the angular distribution of sputtered species in SIO-113 growth. This research provides a unique understanding of how these factors interact to explain the observed phenomena, presenting valuable insights for future improvements in this field.

## 2. Materials and Methods

### 2.1. Film Growth

Epitaxially stabilized perovskite-like SrIrO_3_ films have been prepared by using RF magnetron sputtering on top of (001)-SrTiO_3_ (STO) single crystalline substrates from a non-stoichiometric iridate target [9]. Details of target preparation are described in the Appendix A. Films were deposited in the range of RT-900 °C in a pure oxygen atmosphere between 6 and 140 mTorr. STO substrates were treated before deposition in order to achieve atomically smooth surfaces of TiO_2_ terraces [9]. The course of treatment involved a 10 min ultrasonic leaching in deionized water and a 2 h annealing at 1000 °C [29]. X-ray reflectometry (Siemens D-5000 diffractometer) was used to determine the thickness of SIO-113 samples and to calibrate the growth rate. The SIO films’ thickness used in this work were in the range of 20–30 nm.

### 2.2. Structural and Physical Characterization

Further X-ray measurements were used to evaluate the structural properties of the SIO-113 films. Phase purity was determined from standard θ–2θ diffraction patterns (Siemens D-5000, Aubery, TX, USA), while reciprocal space maps around (103) peaks were recorded by using a Bruker D8-Discover diffractometer (Bruker, Billeria, MA, USA) to assess the epitaxial quality and orientation of the films.

The surface quality of the samples was studied with atomic force microscopy (MFP-3D AFM Asylum Research, Goleta, CA, USA). Topographic images were recorded in tapping mode by using Sb-doped Si probes (NCHV-A from Bruker, Billeria, MA, USA). The surface roughness was calculated from the images as a root mean square (rms) of the height distribution. Cation stoichiometry was analyzed by electron probe microanalysis with commercial JXA-8230 equipment from JEOL (JEOL Ltd., Tokyo, Japan). X-ray photoelectron spectroscopy (XPS) measurements were performed with Phoibos 150 equipment from SPECS (SPECS, Berlin, Germany). To preclude the possible presence of contaminants, an overview scan in the range from 0 to 1380 eV was performed firstly. Afterwards, the most relevant peaks were studied in detail to obtain information about the surface composition. In general, these peaks were 3d-strontium, 4f-iridium, 1s-carbon, 1s-oxygen and 2p-titanium. To determine the films’ stoichiometry, measurements were taken at different areas of the sample. From these experimental deviations, we estimate an average error of 15% in the determination of the stoichiometric ratio of metallic cations.

The temperature dependence of the resistivity was measured by using a standard four-probe configuration in a PPMS system from Quantum Design between 10 K and 300 K during warming (3 K/min). Measurements were performed along a [100] crystallographic axis in 400 μm long × 100 μm wide tracks patterned by UV lithography and physical etching. Metallic Pt contacts were deposited by sputtering through a suitable shadow mask, and then samples were contacted by using aluminum wire. Electrical current was kept below 50 μA in all the measurements to avoid local heating.

## 3. Results and Discussion

### 3.1. Influence of Growth Conditions

Achieving high-quality stoichiometric oxide films typically requires precise adjustments in the growth conditions. The most straightforward method involves studying the temperature–oxygen phase diagram to pinpoint the specific zone where the preferred oxide phase is thermodynamically favored. In pursuit of this objective, we conducted the growth of a sequence of strontium iridate thin films. These films were grown at various nominal substrate temperatures spanning from 500 °C to 900 °C (the maximum operational temperature within our system) and under oxygen partial pressures ranging between 6 mTorr and 140 mTorr. Films were evaluated by studying their surface roughness and X-ray diffraction patterns. Figure 1 shows the impact of the growth temperature on the microstructure of the films.

The AFM images in Figure 1 (top row) expose a clear evolution from a flat film at 500 °C to a rough film at 700 °C and then a progressive flattening of the surface up to 900 °C. This roughness evolution may be tracked in the right panel of Figure 1. The RMS roughness reached a minimum value below 0.2 nm, making it appropriate for applications with stringent interfacial quality requirements. At the lowest growth temperature of 500 °C, only diffraction peaks corresponding to the STO substrate were detected, thus indicating the absence of epitaxial iridate films (see bottom panels in Figure 1). With increasing T, a diffraction peak develops that can be easily identified as the (002) reflection of the perovskite-like n = ∝ SIO-113 phase [15,23]. The peak intensity rises with the growth temperature, indicating enhanced crystallinity. A small shift in the peak was observed that may be attributed to slight differences in oxygen content between the samples. The temperature-dependent evolution of the film’s microstructure suggests that the primary effect of the growth temperature is to boost the diffusion of adatoms on the substrate surface, thereby promoting a layer-by-layer growth mechanism.

A similar trend was observed when varying the background oxygen partial pressure during growth (see the Appendix A). It is believed that higher oxygen pressures promote more collisions between species, leading to a better thermalization of the atoms and decreasing the growth rate [15,30]. Unexpectedly, in our case, the thickness of the films has no significant dependence on background pressure. X-ray diffraction patterns (see Appendix A) again evidence the exclusive formation of the n = ∝ SIO-113 phase as only 00l peaks of this phase have been observed. Its position at 2θ = 44.75° corresponds to an out-of-plane cell parameter (assuming pseudocubic phase) c = 0.405 nm. Reciprocal space maps (not shown, see Ref. [9]) confirmed that, in the range of thickness studied in this work, SIO films are always fully strained with the STO underlying substrate (a_STO_ = 0.3905 nm) [9,10,20]. The observed cell parameters point towards an elastic behavior solely attributed to the in-plane compressive strain imposed by the substrate, consistent with prior reports [9].

Our results in Figure 1 suggest that, in spite of the non-stoichiometry of the target, sputtering processes favor the nucleation of the perovskite-like phase of strontium iridate. We have not been able to find any region in the temperature–oxygen phase diagram where n = 1 or where any other Ruddlesden–Popper phase could be identified (from X-ray experiments). These findings are in clear contrast to those reported by other groups using different growth methods. For example, it was demonstrated early that just by modifying thermodynamic growth conditions for different film phases, including the n = 1 phase, Sr_2_IrO_4_ could be grown from the very same single target [24,25]. In those works, the perovskite-like n = ∞ phase just occurred at the low temperature region of the phase diagram with a crossover towards the n = 1 phase when temperature was raised (at 100 mTorr mixing of phases is observed as low as 700 °C) [24]. It is noteworthy to mention that, in our scenario, additional experiments involving different background atmospheres, such as a mixture of argon and oxygen to maintain a low oxygen partial pressure while keeping the total pressure constant, resulted in phase segregation and the emergence of metallic iridium at the film surface [8]. In order to elucidate this deviation in our experiments from the commonly reported findings in the literature and gain insights into the various nucleation processes, a more detailed study is thus deserved.

### 3.2. Film Stoichiometry

Since films are grown from a non-stoichiometric (strontium-rich) target, it is natural to analyze first the stoichiometry of the films. A usual technique to verify the final film composition is microprobe measurements. However, as film and substrate have two elements in common, oxygen and strontium, it is difficult to extract reliable information from the measurements. For this reason, we have grown some samples on top of NGO (NdGaO_3_) substrates under the same conditions. The X-ray diffraction patterns were equal to those obtained from STO, thus suggesting that, even if chemistry and cell parameters are different, the SIO-113 growth process could be considered similar. Microprobe experiments showed that there is an excess of strontium in relation to iridium in the films with an atomic relationship close to [Sr]/[Ir] = 1.3 (we should remind readers that the nominal target ratio is 2). Since our focus lies on understanding this cationic relationship, employing X-ray photoelectron spectroscopy (XPS) emerges as a suitable method for gaining deeper insights [31]. Our approach involved a set of XPS experiments crafted to unveil the stoichiometric distribution across the film’s depth. Initially, distinct square regions on a uniform film were delineated via photolithography. These selected areas were subsequently subjected to ion milling with varying durations while maintaining a consistent attack rate. Following this, the depth of the milled regions was measured using AFM profiling to establish the ion milling rate (0.2 nm/s in our case). Then, a full SIO-113 film was etched at successive increased times. After each milling proccess, XPS experiments were performed to obtain information on the stoichiometry profile. The results are presented in Figure 2.

As it can be observed in the graph, the atomic oxygen (blue triangles) takes values around 60% for the whole film as well as for the substrate, which is in good agreement with the nominal value in perovskites. On the other hand, strontium cations (red circles) seem to have a higher concentration at the surface of the film (>25%), and after the first nanometers they decay to a constant value slightly above the nominal 20%. Since both the film and the substrate have the same nominal value for Sr, once the interphase is crossed, the Atomic% of the Sr remains constant. Iridium cations (black squares) represent only the 13% of the atoms at the surface, compensating in this way for the overpopulation of Sr. Beyond the surface, the value of the Ir cations increases and it is stabilized at values of 16–19% through the main part of the thickness, which is slightly below the nominal value. Close to the interface with the substrate, iridium starts to decrease smoothly first. At the same time, the titanium signal slowly starts to rise, indicating that any possible interdiffussion is limited to the very first atomic layers of the films. However, we should note that the XPS-probed volume extends to a certain depth and that Ti signals may appear even before the whole SrIrO_3_ film has been etched. Thus, the interface quality can not be precisely assesed from these experiments. In summary, films exhibit a constant cationic ratio through most of their bulk volume while being strontium-rich (iridium-deficient) close to the free surface. By integrating the whole profile, an overall cationic ratio of [Sr]/[Ir] = 1.30 has been obtained, in very good agreement with the above microprobe measurements of SIO films grown on NGO substrates. Since no titanium diffuses into the bulk of the film beyond the initial layers, the existence of a certain quantity of iridium vacancies could be inferred.

### 3.3. Sputtering Processes

Physical vapor deposition techniques involve processes of a non-equilibrium nature. It is a fact that cation composition could be affected by kinetic growth mechanisms during the transfer of materials from a multielement target as previously observed also in PLD films [25,26,32]. Thus, to understand the results from the precedent section, we have concentrated on two possible mechanisms that may influence cation stoichiometry. First, we will focus on backsputtered processes and then we will analyze the angular distributions of sputtered atoms.

#### 3.3.1. Backsputtering Phenomenon

It is known that energetic O^2−^ particles generated in the plasma bombard the growing film, causing the reemission of adatoms, therefore reducing their sticking probabilities [4,30]. This backsputtering phenomena may strongly modify the final cation composition and several processing parameters as source power or target distance play an important role in controlling it. We will, however, focus our interest on the less studied role of substrate temperature. Note that other mechanisms, such as, for example, different flight times for the various atomic species, are also important in the amount of material that lands on the substrate surface. However, as there is no way to discriminate between all those processes, for the sake of simplicity, we are just referring to them under the general term of backsputtering.

Figure 3 depicts the changes in thickness and cation composition of the sputtered iridate films obtained from an identical target when deposited at different substrate temperatures by PLD (black) and by sputtering (orange). Figure 3 evidences that PLD and sputtered films follow completely different trends when the temperature of the substrate is increased. On one hand, PLD films do not show any clear correlation between temperature and thickness (Figure 3a) as the observed small fluctuations may be attributed to experimental variability. In contrast, films deposited by magnetron sputtering clearly show a reduction in thickness as temperature is increased. It is worth mentioning that films deposited at 900 °C are nearly an order of magnitude thinner than films deposited at room temperature. As the experiments were performed under identical working target-to-substrate distance and power source, this huge difference in deposited material can only be attributed to backsputtering processes. This implies that, in sputtering, substrate temperature decreases the binding energy of the adatoms at the surface, and the energy threshold for their reemission is more easily achieved. At a simple glance this phenomenon looks hard to master. Furthermore, the proposition that atomic mass could exert an influence has been substantiated through an examination of the stoichiometry of the deposited films. Figure 3b illustrates that while PLD films accurately mirror the composition of the target in the deposited film, sputtered films exhibit a more notable dependence on this factor. Surprisingly, at low temperatures, films are highly deficient in strontium with [Sr]/[Ir] = 0.5. However, above 700 °C, the cation ratio approaches 1.0, and then it continues to increase up to a maximum of 1.3. This indicates that strontium is very prone to backscattering processes and only the formation of the iridate phase starting at 700 °C (from results of Figure 1) helps to stop this loss of strontium. Nevertheless, in the whole range of measured temperatures, the [Sr]/[Ir] ratio never arrives at the nominal value of the target, i.e., [Sr]/[Ir] = 2.

#### 3.3.2. Angular Distribution

The other process that may influence the deposited films deals with the anisotropy of the flying atoms once extracted from the target. Given the distinct atomic mass and vapor pressure, it is reasonable to assume that sputtered strontium and iridium cations may exhibit a different angular distribution [26,30,33]. One easy way to study this distribution is to perform off-axis sputtering experiments. In this configuration, the substrate is placed out of the axis of the target. With slight variations in this configuration, off-axis sputtering is a usual technique to achieve high-quality thin films by reducing backsputtering and, in consequence, controlling stoichiometry.

We have grown a series of films at different distances, Δx, from the normal axis of the target, as indicated in the image in Figure 4a. The results of the depositions at room temperature (RT) and at 900 °C are depicted in Figure 4c,d. In Figure 4c, it can be observed that when deposited at RT, the thickness of the film rapidly decreases with Δx. The most relevant variation is found between 0 and 25 mm, where the drop in thickness is almost an order of magnitude. After this, the thickness follows a smooth decrease tendency with Δx. On the other hand, at high temperature (900 °C), the thickness of the films remains constant, and only above 25 mm a reduction in thickness can be observed. Interestingly, beyond 25 mm, both behaviors (at room temperature and at 900 °C) appear to converge, with room-temperature films being marginally thicker; however, the overall trend remains consistent for both curves. These findings corroborate our earlier analysis, emphasizing the substantial impact of backsputtering mechanisms on the growth of SIO-113 films. In a very simple model, we may assume that the values obtained at room temperature are a good measure of the material landing at the substrate and the differences observed when growing at higher temperature are mostly due to backsputtering. With this in mind, Figure 4c indicates that when the flux of energetic species is high enough (small Δx, i.e., close to on-axis), backscattering is very important at high temperatures (a huge amount of material is removed from the substrate). On the other hand, when the flux of energetic species is lowered (by moving apart from the on-axis configuration, i.e., high Δx), backsputtering, although still present, is severely reduced. Let us now investigate the influence of off-axis configuration on the cation stoichiometry. Figure 4d shows the dependence of the cation ratio [Sr]/[Ir] as a function of off-axis distance, Δx, for films grown at RT and 900 °C. At room temperature, on-axis films are strongly strontium-deficient (or iridium-rich), as previously commented with a cation ratio of [Sr]/[Ir] = 0.5. This is also true for off-axis films up to Δx~10 mm. Above that, strontium content monotonically increases to reach the nominal target ratio [Sr]/[Ir] = 2 at around 25 mm. For larger values of Δx, the strontium over iridium ratio continues to rise, up to values of [Sr]/[Ir] = 2.5 around Δx~30 mm. Some kind of equilibrium is achieved at this distance and compositional ratio is unchanged in the remaining samples. These results may be explained assuming that the angular distribution of the ejected atoms is different for strontium and iridium. Iridium-rich on-axis film suggests that iridium plasma is more concentrated along the on-axis configuration, while strontium-rich films for the extreme off-axis configuration lead to a wider distribution of strontium. The schema of Figure 4b depicts a very simple sketch of the possible angular plasma configurations for both sputtered species. At high temperatures (900 °C), the situation is quite different. On-axis films and up to Δx~20 mm off-axis films exhibit a constant ratio composition of [Sr]/[Ir] = 1.3. Beyond that specific distance, there is a noticeable shift in stoichiometry, characterized by a clear strontium enrichment. To explain this behavior, it is essential to consider the role of the three underlying mechanisms. First, RT results suggest a different angular distribution of sputtered Sr vs. Ir atoms. Second, backsputtering is more important at high temperatures and could be different for both species. Third, at 900 °C, as the perovskite-like n = ∞ phase is formed, there are additional mechanisms for attaching the strontium ions to the substrate. The results in Figure 4d are just evidence of the delicate balance between these effects and the fine-tuning needed to obtain oxide thin films of the right stoichiometry and properties.

The iridate films grown using “off-axis” conditions only revealed well-crystalized epitaxial phases when high temperature was employed. Figure 5 shows the X-ray θ–2θ diffraction patterns obtained for this series. The figure shows that films grown up to 2.4 cm exhibit similar X-ray diffraction patterns with peaks corresponding only to the perovskite n = ∞ phase SrIrO_3_. In this range, the overall stoichiometry was unchanged, close to [Sr]/[Ir] = 1.3–1.5, and selective nucleation of the perovskite phase followed an identical trend. Small differences in the peak position for samples below 2.4 cm may be attributed to slightly different oxygen content, as in the case of Figure 1. However, for the off-axis (Δx = 4 cm) sample with a stoichiometric ratio [Sr]/[Ir] = 2.8, the diffraction peak moved from 2θ = 44.7° towards 2θ = 44.1°, still far away from the expected positions of both the n = 1 (Sr_2_IrO_4_) and n = 2 phases (Sr_3_Ir_2_O_7_). This shifting cannot be only attributed to oxygen vacancies, and, instead, a plausible scenario arises where the strontium excess induces an extra chemical strain that strongly distorts the perovskite phase. Furthermore, given the absence of additional reflections, it is not possible to attribute this peak to any phase other than a considerably distorted n = ∞ phase. Therefore, despite having a stoichiometrically sufficient amount of strontium for the growth of another Ruddlesden–Popper phase, epitaxial stabilization takes precedence over alternative growth mechanisms. This results in the nucleation of the n = ∞ phase and the accommodation, if needed, of significant variations in stoichiometry (such as an excess of strontium). To fully understand the behavior of strongly off-stoichiometric film grown off-axis, and, in particular, to explore the chemically induced strain in these RP series as a strategy for improved properties [34], a complete microscopic study would be required, which, although interesting, is beyond the scope of this work.

### 3.4. Thin-Film Stability

It has been previously reported that strontium iridate films are prone to aging, and the surface is rapidly degraded when in contact with air [35]. This raises a severe concern for their use in several applications relying on surface quality, such as, for example, spintronics. To assess the stability of the films, we have studied the evolution of electrical resistivity over several months. Figure 6 shows the typical results for an as-grown film and after being stored in a low-humidity atmosphere. Figure 6a shows that resistivity remains unaltered after a few months of storage (less than 0.3% of variation). The resistivity of the film exhibits a metallic regime in a wide range of temperature and, at very low temperatures, a small upturn can be observed, with results that are similar to those previously reported [9,36,37,38,39,40]. In addition, the cation composition was verified and found to remain unchanged. Nevertheless, there was an observed progressive alteration in the film’s surface over time. Figure 6b,c show the AFM images of the very same film just after growth (b) and after 5 months of storage (c). The initial image captured after growth displays a flat morphology. However, after a span of 5 months, the film undergoes a transformation, becoming uniformly coated with small particles that notably increase the film’s roughness. Since the composition of the surface is unaltered with time, these particles probably arise from the atmospheric carbon/organic contamination. However, this degradation is purely superficial and it does not alter the macroscopic measurements or intrinsic properties of the film.

## 4. Conclusions

The study presents a unique analysis of the growth of Ruddlesden–Popper strontium iridate thin films via magnetron sputtering. In contrast with previous reports, our results make evident the formation of the perovskite-like n = ∞ SrIrO_3_ phase despite using a non-stoichiometric target. A careful analysis of the cation stoichiometry of the sputtered films as a function of various parameters allows us to demonstrate the existence of remarkably different angular sputter distributions for strontium and iridium atoms, which could be of great relevance for the preparation of Ruddlesden–Popper strontium iridate thin films. Additionally, the study identifies an important backsputtering phenomenon at high temperatures, attributed to reactive oxygen ions. By elucidating the interplay between these mechanisms, our results demonstrate the prevalence of kinetic effects over thermodynamic conditions in sputtering growth, offering new opportunities to control and fine-tune cation stoichiometry in oxide thin films. The stability of the oxide films over extended periods is also demonstrated, underscoring the practical relevance of our findings.

## Figures and Tables

**Figure 1 nanomaterials-14-00242-f001:**
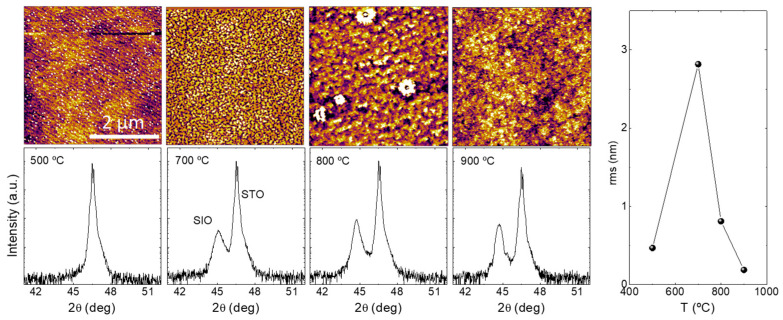
**Top** panels: AFM surface images. **Bottom** panels: X-ray diffraction patterns. **Right** panel: evolution of rms roughness of a series of SrIrO_3_ thin films deposited at different temperatures and oxygen pressure of 140 mTorr.

**Figure 2 nanomaterials-14-00242-f002:**
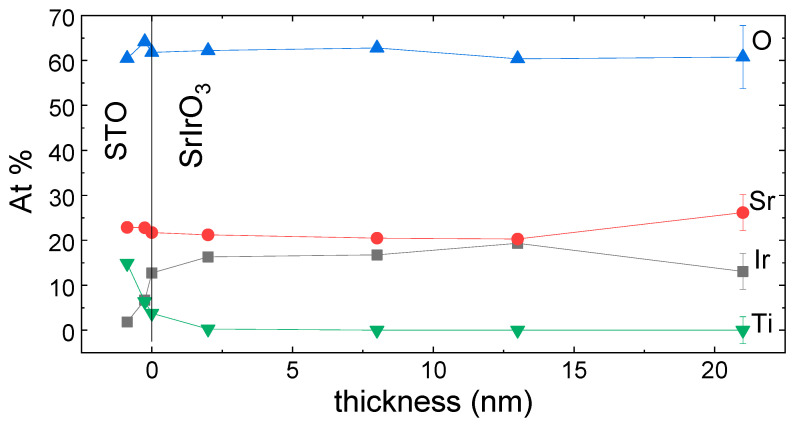
XPS percentage atomic profile of the different ions present in the SrIrO_3_ films. Data are represented as a function of the film thickness from the interphase with the substrate.

**Figure 3 nanomaterials-14-00242-f003:**
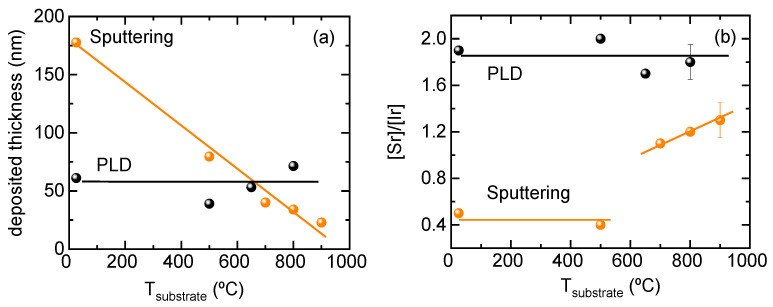
Variations in (**a**) film thickness and (**b**) cation stoichiometry with the nominal substrate temperature for films grown by PLD and sputtering. The iridate films were grown by PLD and sputtering.

**Figure 4 nanomaterials-14-00242-f004:**
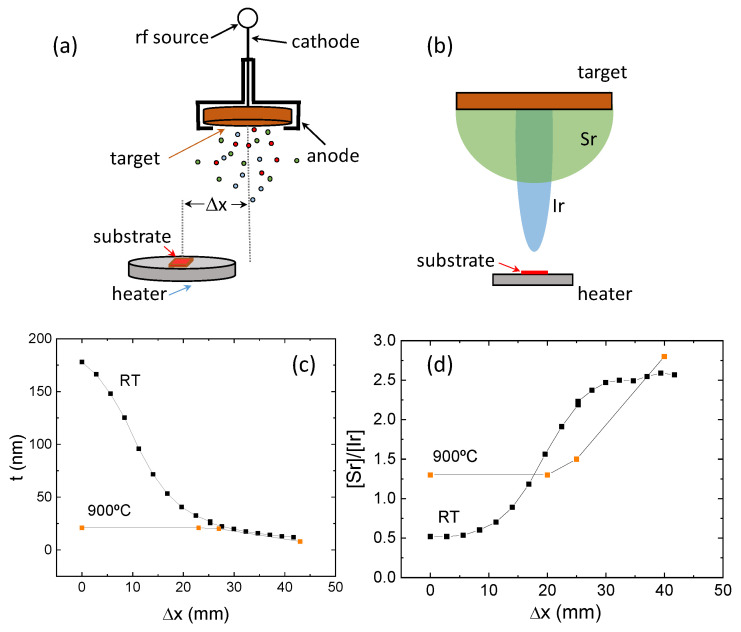
(**a**) Scheme showing the sputtering setup used in this work. Δx measures the displacement from the on-axis target–substrate configuration. (**b**) Sketch of the angular distribution of the sputtered cationic species. (**c**) Thickness of the films after 1h of deposition at the same power 20 W at different Δx at room temperature (black dots) and 900 °C (orange dots). (**d**) Cation composition [Sr]/[Ir] at different Δx from the same samples as in (**c**).

**Figure 5 nanomaterials-14-00242-f005:**
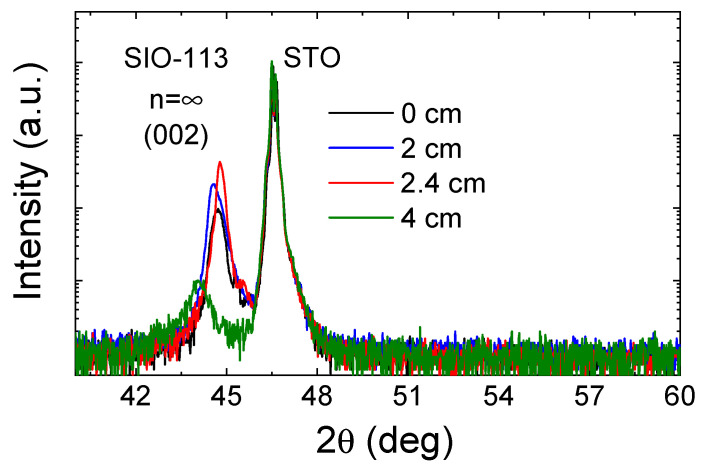
θ–2θ measurements of the iridate films deposited at 900 °C at different off-axis Δx positions.

**Figure 6 nanomaterials-14-00242-f006:**
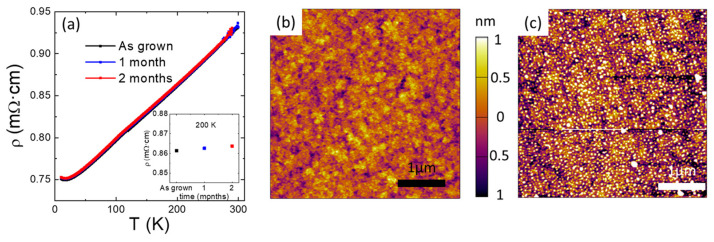
(**a**) Aging of the temperature dependence of the electrical resistivity of a SrIrO_3_ film. Inset shows the dependence on aging time of the room temperature resistivity. (**b**) AFM image of a SrIrO_3_ film just after deposition. (**c**) AFM image of the same film as in (**b**) after a period of 5 months. Films were stored under inert atmosphere.

## Data Availability

Data are contained within the article and Appendix A.

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
