# Peer review of "Evaluation of Sputtering Processes in Strontium Iridate Thin Films"

_nanomaterials, 2024, doi:10.3390/nano14030242_

Round 1

Reviewer 1 Report

Comments and Suggestions for Authors
I think that this work can be considered for publication after revision.
1.       The authors should formulate very briefly in the introduction the novelty of the work with respect to the other works in the field.
2.       What is the thickness of the strontium iridate thin films? Why did the author choose this thickness in this paper?
3. The conclusion is just a list of observations like for a technical report. Please add the science including for cause of the effects in order to advance the understanding of the studied phenomena compared to what is already known or could be expected from literature.

4.    The accuracy of the measurements of the technology should be presented.

5 . In summary, kindly include the advantage of the present study over other studies.  

Reviewer 2 Report

Comments and Suggestions for Authors

The authors investigate the growth of strontium iridate thin films by magnetron sputtering. In their studied conditions, 113-type SrIrO3 phase is exclusively formed, which can be attributed to the different angular sputtering distribution and backsputtering phenomenon. However, some necessary revisions are required to improve the quality of the manuscript before acceptance.

1.      For better understanding, the authors should provide a schematic diagram for the magnetron sputtering of epitaxial thin films.

2.      We can observe obvious peak shift in the XRD patterns in Figures 1 and 5. Please analyze this by combining structural strain, which is also an important parameter for the study of thin films. The authors can refer to this work 10.1063/5.0083059 for the analysis of structural strain.

3.      The electronic structural properties of the materials are missing in this version. The XPS for Sr, Ir and O elements should be analyzed, especially for O 1s XPS, which can be fitted and extract the detailed oxygen species in the materials. The authors can refer to this work 10.1038/s41467-020-17108-5 for this point.

4.      TEM images should be performed to help analyze the structural information.

5.      Please explain why SrTiO3 substrate is chosen and whether different substrates could affect the films.

Comments on the Quality of English Language

Minor editing of English language required

Reviewer 3 Report

Comments and Suggestions for Authors

The work presented the growth of epitaxial thin films of the Ruddlesden-Popper (RP) family of strontium iridates by magnetron sputtering. The research highlights the prevalence of kinetic mechanisms over thermodynamic ones in the growth process. The results are presented and discussed in a consistent manner. However, I would recommend supporting the conclusions with more bibliographical references in general. The figures need more resolution, especially the AFM ones. I would also recommend including the results, particularly for roughness, for thin films deposited at 600ºC in order to maintain the range of temperature increases.
